# New Insights into the Diagnosis and Treatment of Hepatocellular Carcinoma

**DOI:** 10.3390/biomedicines13051244

**Published:** 2025-05-20

**Authors:** Chengbo Li, Bingjiu Lu, Baocheng Deng

**Affiliations:** 1Department of Infectious Diseases, The First Affiliated Hospital of China Medical University, Shenyang 110001, China; cmu047401@sina.com; 2Department of Hepatology, Affiliated Hospital of Liaoning University of Traditional Chinese Medicine, Shenyang 110032, China

**Keywords:** hepatocellular carcinoma, liquid biopsy, immunotherapy, artificial intelligence

## Abstract

Hepatocellular carcinoma remains one of the leading contributors to global cancer mortality, frequently stemming from chronic liver conditions, such as viral hepatitis, non-alcoholic fatty liver disease, and alcohol-induced cirrhosis. While antiviral treatments have made significant strides, the rising prevalence of hepatocellular carcinoma linked to non-infectious causes underscores the pressing demand for more effective diagnostic tools and therapeutic interventions. Advances in imaging and liquid biopsy technologies have facilitated early detection and diagnosis, and treatment strategies are diversifying to include immune checkpoint inhibitors, tyrosine kinase inhibitors, and interventional therapies. Translational therapies for advanced hepatocellular carcinoma have improved surgical opportunities and patient survival. Artificial intelligence has played a transformative role in the diagnosis and treatment of hepatocellular carcinoma, in terms of image analysis, histopathologic classification, drug development, and targeted therapy. The future of hepatocellular carcinoma treatment lies in precision oncology and the collaboration of multidisciplinary teams, as well as in early detection. The ultimate goal is to keep patients alive longer and reduce the global burden of this complex malignancy.

## 1. Introduction

Hepatocellular carcinoma (HCC) is one of the deadliest forms of cancer worldwide, claiming countless lives each year. The development of the disease is closely tied to chronic liver diseases, such as Hepatitis B and C, non-alcoholic fatty liver disease (NAFLD), and cirrhosis linked to alcohol abuse [1,2]. While antiviral therapies have extended the lives of patients with hepatitis and cirrhosis and even curbed liver cancer rates in some cases, there is a growing concern. The prevalence of HCC stemming from non-infectious liver diseases is on the rise, solidifying its status as a pressing worldwide health challenge. Ranking as the sixth most common malignant tumor globally, HCC stands as the third deadliest cancer worldwide. Despite strides in diagnostic and therapeutic approaches, the outlook for HCC patients remains grim, with a mere 20% survival rate at the five-year mark [3,4]. To turn the tide, prioritizing screening, early detection, and prompt intervention for at-risk groups is essential to improving survival outcomes.

HCC has a complex and varied pathogenesis, necessitating diagnostic methods to adapt to the underlying causative factors and endemic areas. Currently, serological tests and imaging techniques are the most common methods for early diagnosis of HCC. The deepening of research on biomarkers of HCC, such as genomic biomarkers, epigenetic inheritance, and metabolites, as well as the deepening of research on liquid biopsy and multi-omics combined models, has increased the possibility of early diagnosis of HCC. Various types of immune and targeted drugs have been successfully introduced based on the immune escape mechanism related to HCC, and the integration of conversion and interventional therapies for patients with non-resectable HCC has increased the odds of surgery and notably prolonged tumor-related survival. This offers a glimmer of hope for the treatment of HCC. In recent years, artificial intelligence (AI) has had remarkable growth. It has been extensively utilized in imaging, pathology, disease management, and drug development. AI has revolutionized the traditional medical paradigm. There is a strong belief that AI will emerge as an effective means of combating HCC in the future [5].

## 2. Pathophysiology of HCC

The precise cellular origin of HCC remains a topic of ongoing debate. Similar to other carcinomas, HCC may arise from liver stem cells, a transient amplifying cell population, or even fully differentiated hepatocytes. The development of HCC is a multifaceted, multi-step process driven by a combination of genetic susceptibility, viral and non-viral risk factors, and intricate interplay within the cellular microenvironment, including immune cells. Chronic liver disease further compounds these effects, creating a fertile ground for malignant transformation. A critical factor in HCC progression is the dynamic tumor microenvironment. If HCC tumor cells fail to effectively display antigens and evade immune detection—or if the microenvironment is saturated with cells and soluble factors that suppress or neutralize cytotoxic T lymphocytes—the tumor can successfully dodge the host’s immune defenses, allowing it to thrive unchecked. The initial stage in triggering a tumor-specific T-cell response hinges on the expression of tumor antigens. During hepatocarcinogenesis, a spontaneous immune response can be activated due to the abnormal expression of oncofetal and cancer genes. Conversely, genomic mutations that arise as HCC progresses can result in amino acid alterations, ultimately resulting in neoantigens [6,7]. Tumor mutational burden (TMB) is frequently employed as a surrogate measure for the abundance of neoantigens because the likelihood of identifying T lymphocytes specific to these neoantigens correlates with TMB. Advances in sequencing technology, particularly next-generation sequencing (NGS), have enabled the comprehensive mapping of numerous tumor mutation profiles [8,9].

HCC is characterized by a diverse array of soluble mediators that play a pivotal role in modulating anti-tumor immune responses. For example, TGF-β exerts a suppressive effect on anti-tumor immunity at multiple stages. This cytokine not only hampers the activation of dendritic cells (DCs) but also impairs the functionality of T and natural killer (NK) cells. Similarly, vascular endothelial growth factor (VEGF), which is secreted by both tumor cells and the surrounding stroma, not only drives tumor angiogenesis but also undermines the antigen-presenting capabilities of DCs and the stimulatory potential of T cells. The anti-tumor efficacy of VEGF inhibitors can, in part, be attributed to their ability to counteract these immune-suppressive mechanisms [10]. IFN-γ is a cytokine mainly produced by activated T lymphocytes and NK cells. It plays a significant role in immune regulation, antiviral defense, anti-tumor response, and resistance to intracellular pathogens. In patients with HCC, defects in the interferon signaling pathway, HCC secret inhibitory molecules (e.g., IL-10, TGF-β), and reduced serum levels of IFNγ are closely linked to more advanced disease stages and worse prognoses [11]. Genetic alterations are associated with the occurrence and development of HCC, such as proto-oncogenes, tumor suppressor genes, and important genes involved in the cell proliferation cycle, apoptosis, and cell differentiation. Gene deletion, insertion, and translocation are also closely related to the occurrence of HCC [12].

Multiple cellular pathways play significant roles in the occurrence of HCC, such as the Ras/Raf/MAPK pathway [13]. Some of these pathways influence the progression of HCC by regulating angiogenesis and autophagy, and cell apoptosis, proliferation, metastasis, and invasion also participate in the occurrence and development of HCC [14,15]. As a basic form of programmed cell death, apoptosis is a powerful weapon for the immune system to eliminate virus-infected cells or cancerous cells. In most cancers, the regulation of apoptosis is defective, and the dysregulation of apoptosis can lead to the loss of suicide ability of cancer cells [16].

Research on these mechanisms provides a theoretical basis for immunotherapy and targeted therapy for HCC. Many studies have found that autophagy, an evolutionary process responsible for the degradation of cellular substances, is closely associated with the acquisition and maintenance of drug resistance in HCC [15]. Some studies related to ferroptosis have also brought some glimmers of hope to current research on HCC. Ferroptosis is a type of iron-dependent regulated cell death triggered by the toxic accumulation of lipid peroxides on the cell membrane. This provides new opportunities for HCC therapy. The induction of ferroptosis can kill drug-resistant cancer cells [17,18].

All of these studies have confirmed that immune escape is an important research direction in the treatment of HCC.

## 3. Early Diagnosis of HCC

### 3.1. Imaging

HCC can be diagnosed by non-invasive examination, such as computed tomography (CT) or magnetic resonance imaging (MRI), owing to its characteristic radiographic features, which include arterial hyperenhancement, venous washout, and capsule enhancement [19]. Alpha-fetoprotein (AFP), AFP-L3 (alpha-fetoprotein heterogeneity-L3), and des-γ-carboxy prothrombin (DCP) are used as auxiliary indicators for HCC diagnosis.

LI-RADS is a standardized framework for interpreting liver imaging in high-risk patients (cirrhosis, chronic hepatitis B, or prior HCC) designed to improve consistency in diagnosing HCC and other liver lesions. It reduces variability in imaging interpretation, it is evidence-based with regular updates (e.g., LI-RADS 2018), and integrates seamlessly into multidisciplinary HCC care through the disposal of high-risk groups.It is necessary, for early diagnosis of HCC Early diagnosis of HCC is and functional imaging and the use of hepatobiliary-specific MRI contrast agents, such as gadoxetic acid disodium (Gd-EOB-DTPA), have revolutionized HCC diagnosis by enabling the precise detection of small and early-stage HCC. Contrast-enhanced ultrasound (CEUS) enhances blood flow signals through microbubble contrast agents and combines real-time dynamic imaging. It demonstrates unique advantages in the diagnosis, staging, treatment guidance, and efficacy evaluation of HCC. Currently, it has been incorporated into LI-RADS as an important tool for HCC diagnosis [20]. New microbubble contrast agents (such as Sonazoid) have the characteristic of being taken up by Kupffer cells. Due to the lack of Kupffer cells in HCC, it shows a negative image and increases the detection rate of atypical lesions of HCC [21].

The advancement of CT technology has also greatly enhanced the rate of early detection of liver cancer. Multidetector CT (such as 256/320-slice CT) enables sub-millimeter thin-layer scanning (0.5–0.625 mm), which enhances the detection rate of small lesions (<1 cm) [22]. Special CT technology (e.g., Spectral CT, Perfusion CT) not only increases the detection rate of HCC but also can be used to evaluate the therapeutic response and prognosis of HCC [23,24,25].

### 3.2. Liquid Biopsy and New Biomarkers

#### 3.2.1. New Biomarkers

AFP, vitamin K deficiency, antagonist-II induced protein (PIVKA-II) (DCP), and AFP L3 are all traditional serum tumor markers; AFP is often used as a serologic marker for the diagnosis of HCC. Some patients with HCC have normal or low levels of AFP even when the disease is advanced. A total of 15–30% of patients with advanced HCC have normal serum AFP levels, and the use of vitamin K-containing drugs may result in lower PIVKA-II values. We found that vitamin K antagonists and diseases causing vitamin K deficiency (e.g., biliary obstruction or cholestasis) may lead to elevated PIVKA-II values [26]. Increased PIVKA-II concentrations may occur in patients with renal insufficiency. Drugs containing vitamin K analogs may also have caused bias. To avoid the limitations of traditional tumor markers and achieve early diagnosis of HCC, the concept of new tumor markers has been proposed: GPC3 is a cell surface proteoglycan, similar to AFP, which is usually undetectable in healthy adult livers and can be observed only in fetal livers. Glypican-3 (GPC3) can be used as a key complementary biomarker for AFP-negative HCC, and it significantly improves the sensitivity and specificity for the diagnosis of HCC [27]. Osteobridging protein (OPN) is another highly promising serologic biomarker for the diagnosis of HCC; we have found that it demonstrates diagnostic advantages in small HCC and cases of AFP-negative HCC [28]. Golgi protein 73 (GP73) levels are elevated in patients with HCC, and GP73 possesses superior sensitivity and specificity to AFP in predicting early-stage HCC; it is also a promising therapeutic target for HCC. GP73 overexpression in carcinoma cells enhances the production of VEGF and pro-mitotic signaling in vascular endothelial cells, thereby promoting angiogenesis in the tumor microenvironment (TME) [29]. Additionally, the level of heat shock protein 70 (HSP70) helps differentiate early-stage precancerous HCC from progressive HCC. Researchers have found that the combined detection of HSP70 and GPC3 significantly improves the sensitivity and specificity of HCC diagnosis [30]. The integration of multi-omics technologies and AI analysis is expected to further optimize early screening and personalized treatment strategies for HCC (Figure 1).

#### 3.2.2. Liquid Biopsy

Liquid biopsy represents an advanced diagnostic approach that identifies diseases by examining biomarkers in body fluids, such as blood, urine, and cerebrospinal fluid. This innovative method has gained significant traction for both cancer research and clinical applications. Unlike conventional tissue biopsies, which involve invasive procedures, liquid biopsies provide a non-invasive, user-friendly, and real-time monitoring solution, making them indispensable in the realm of precision medicine. It is not far-fetched to predict that this technique will become increasingly pivotal in shaping the future of medical diagnostics.

Circulating tumor cells (CTCs) are tumor cells isolated from the blood. Some CTCs can survive immune attacks and evade destruction. They provide direct evidence of tumor presence and metastatic potential and play a pivotal role in the prognostic assessment of the disease [31]. Circulating tumor DNA (ctDNA) is a tumor-specific DNA fragment present in body fluids, such as blood, which forms part of the free DNA (cfDNA). Tumor cells produce these fragments via apoptosis, necrosis, and active secretion [32]. Virus-derived sequences, single-nucleotide variants, and aberrant methylation patterns can be used to distinguish ctDNA from non-tumor cfDNA [33]. We believe that ctDNA, when combined with advanced technologies such as NGS and AI, has significant potential for early cancer detection and will play a key role in the early screening and therapeutic monitoring of HCC.

MicroRNAs (miRNAs), messenger RNAs (mRNAs), and long-chain non-coding RNAs (lncRNAs) comprise cell-free RNAs (cfRNAs), which can serve as key liquid biopsy biomarkers for cancer detection and surveillance applications. mRNA (cf-mRNA) has been sequenced, and this scientific group discovered and validated a cf-mRNA signature consisting of 10 genes that effectively differentiated HCC patients from hepatitis patients and normal controls, demonstrating the enormous clinical potential of cf-mRNA for tumor diagnosis and subtype classification [34].

Large-scale genomics and genome-wide studies utilizing comprehensive genomic tools have enhanced our knowledge of cancer evolution and heterogeneity. Genome-wide studies using NGS have explored a wide range of genetic and epigenetic alterations associated with liver tumorigenesis; it is also worth noting that epigenetic modifications, including DNA methylation, histone modifications, and deregulated non-coding RNA expression, are strongly associated with hepatocellular carcinogenesis [35]. DNA methylation is a promising biomarker for cancer. Dong Hua et al. indicate that the analysis of a combination of these six methylated genes may be a promising method for the risk assessment and diagnosis of HCC [36]. It is believed that similar studies will bring hope to the diagnosis of HCC.

Extracellular vesicles (EVs) such as exosomes, microvesicles, and apoptotic bodies can also serve as critical tools in liquid biopsy for HCC early detection and diagnosis [37]. Metabolomic and glycosylated protein markers have demonstrated significant potential for the early diagnosis of HCC [38,39].

## 4. Treatment of HCC

The approach to treating HCC requires a multidisciplinary effort, bringing together expertise from hepatology, surgical teams, diagnostic and interventional radiology, oncology, and pathology. For advanced cases, a combination of treatment strategies is typically used. Historically, surgical resection and liver transplantation have been regarded as the sole curative interventions for HCC. Partial hepatectomy, however, remains a feasible option, but only for a specific subset of patients: those with a single tumor, no significant vascular invasion, the absence of cirrhosis, and no clinically evident portal hypertension. Careful patient selection is critical to ensure the best possible outcomes, and transplantation is reserved for patients with early HCC within the adapted version of the Milan criteria [40]. The formulation of an individualized and systematic conversion treatment plan is undoubtedly the most suitable option for patients. Experts have proposed a combined treatment plan of interventional and systemic therapy and have explored the aspects of conversion treatment for advanced HCC.

### 4.1. Immunotherapy for HCC

HCC progression is regulated by the immune system. Various immune mechanisms play crucial roles in the development and progression of HCC and are closely related to prognosis. Immunotherapy may be an effective treatment for HCC, and immunotherapies, such as anti-angiogenic tyrosine kinase inhibitors (TKIs) and immune checkpoint inhibitors (ICIs), have demonstrated potent anti-tumor activity in some patients. Atilizumab, an antibody that inhibits programmed death ligand 1 (PD-L1), in combination with bevacizumab, a VEGF-neutralizing antibody, is a standard first-line treatment for HCC. This therapy has become, or will soon become, the standard first-line treatment option for HCC. New therapeutic approaches, including ICIs, TKIs, monoclonal antibodies (CIs), cytotoxic T lymphocyte-associated antigen 4 (CTLA4), and VEGF inhibitors, have greatly improved the treatment of HCC.

Studies on survival and disease control in patients with HCC have shown that multimodal therapeutic strategies, such as immune checkpoint inhibitors in combination with VEGF inhibitors, TKIs, or other immunotherapies, have the potential to dramatically improve patient prognosis [41]. The academic exploration of novel immune checkpoint inhibitors, such as lymphocyte activation gene 3 (LAG3) and T-cell immunoglobulin mucin 3 (TIM3) inhibitors, has also revealed promising clinical applications [42,43]. The molecular mechanisms of cancer stem cells (CSCs) and tumor-associated growth factor signaling pathways are becoming increasingly clear, and we are striving to develop targeted therapeutic regimens aimed at interfering with tumor proliferation, metastasis, and treatment resistance [44]. A deeper understanding of the molecular mechanisms of tumorigenesis and progression will lead to the development of targeted therapies. These studies are working towards the development of more effective therapies.

### 4.2. Interventional Therapy for HCC

Interventional therapies have emerged as crucial and increasingly popular treatment options for HCC. These approaches effectively control tumor growth, spare healthy liver tissue, and minimize toxic side effects. In the treatment of early-stage HCC, the selection of locoregional therapy depends on factors such as tumor size, the nature of the underlying liver condition, and overall liver function. Broadly speaking, these therapies fall into two categories: percutaneous and intra-arterial. The latter includes a range of procedures, such as bland embolization (TAE), transarterial chemoembolization (TACE), drug-eluting bead chemoembolization (DEB-TACE), selective internal radioembolization therapy (SIRT), and hepatic arterial infusion chemotherapy (HAIC), all of which leverage the arterial blood supply unique to HCC. Recent advancements in interventional therapy have paved the way for innovative directions, particularly the development of new drug-eluting beads (DEBs) that can be loaded with various chemotherapeutic agents. In a porcine model of hepatic arterial embolization, they measured drug plasma levels, conducted histopathology, and compared the results to those of intra-arterial bolus injection of the drug. The use of DEB resulted in a decrease in peak plasma levels. Precision oncology, based on these findings, is potentially one of the most important aspects of present and future developments in interventional oncology. Imageable radiopaque beads were created. During the bead manufacturing process, a radio-absorber such as iodine or barium is integrated into these particles. These beads enable the real-time confirmation of HCC targeting during the procedure. Immunoembolization is another emerging area, as noted by Yamamoto et al., in which granulocyte-macrophage colony-stimulating factor (GM-CSF), a form of systemic immunotherapy, directly injected into the arteries that supply HCC, has shown promising results. This approach not only extends patient survival but also slows the progression of extrahepatic metastases [45]. In recent years, the development of nanoparticle-based formulations for cancer treatment has surged, with many being designed to deliver therapeutic agents. Additionally, there is a growing interest in exploring other therapeutic strategies, such as light, heat, ultrasound, magnetic fields, redox reactions, reactive oxygen species, and systemic therapies combined with transarterial embolization techniques [46]. Integrating embolization methods with percutaneous ablation therapy can further amplify treatment efficacy and boost patient outcomes [47]. As advancements in chemotherapy and nuclear medicine continue, it is anticipated that more refined and effective interventional approaches will become the standard in clinical practice.

### 4.3. Other Therapy for HCC

Conversion therapy refers to the process of transforming initially inoperable liver cancer into operable tumors through systemic or local treatment, thereby providing patients with the opportunity for radical surgery. Combination therapy, on the other hand, involves the synergistic effect of multiple treatment methods to increase the tumor response rate and prolong survival. Zhang et al. demonstrated triple combination therapy with an oral anti-angiogenic drug, programmed death-1 inhibitors, and hepatic arterial infusion chemotherapy. The surgical conversion rate was 60% [48]. Histotripsy is the recently approved non-thermal, non-invasive, focused ultrasound therapy by the FDA. It is applicable to patients with inoperable tumors or those with potential liver cirrhosis who are not suitable for traditional treatments such as surgery or radiotherapy. This kind of treatment can preserve the natural conformation of tumor antigens, thereby inducing a stronger tumor-directed immune response [49,50].

With the continuous development and progress in medical technology, it is believed that new drugs and technologies will be introduced into clinical treatment in the future.

## 5. The Application of AI to HCC

AI has rapidly emerged as a transformative tool across various domains of cancer research, encompassing diagnosis, grading, drug discovery, treatment development, and the prediction of clinical outcomes [51,52] (Figure 2).

### 5.1. AI in HCC Research

Over the past few decades, there has been an explosion of large and complex datasets (containing genomic and molecular data from numerous tissues and individual cells) available to us. To improve the detection and characterization of HCC, the academic community has designed a variety of AI algorithms that integrate multi-omics approaches. Deep learning, often referred to as deep neural networks, relies on a hierarchical structure of hidden layers, where data are processed and features are extracted through a sequential flow of interconnected nodes [51]. AI excels in creating synthetic data, which essentially mimics real-world patterns and characteristics, thereby enhancing target-recognition capabilities. By leveraging AI-driven algorithms to model diverse biological scenarios, researchers have afforded greater flexibility in investigating and dissecting complex phenomena, opening new avenues for exploration and analysis [53,54]. AI has the potential to create synthetic data using existing knowledge and patterns. These data can be used to train AI models and uncover potential therapeutic targets that have been overlooked in the past. and assists in the discovery of therapeutic targets through analysis of the biomedical literature. By mining previous studies, they correlated diseases, genes, and biological processes, quickly identified the biological mechanisms involved in disease development and progression, and identified potential drug targets and biomarkers [55]. Cannon and colleagues demonstrated that by analyzing the connections among approved medications, molecular targets, and therapeutic indications, AI can expedite drug development. This is achieved by connecting the well-established high-confidence targets of existing drugs to novel, uninvestigated diseases [56]. Verifying targets using cell and animal models is a crucial phase in target discovery. This step can reduce the project failure rate and drug development expenses in the pharmaceutical sector. Ren et al. (2023) by leveraging a deep-learning approach, pinpointed cyclin-dependent kinase 20 (CDK20) as a therapeutic target for HCC [57]. Small molecule inhibitors, with the assistance of alpha fold and being finally successfully selected, exhibit selective anti-proliferative effects in HCC cell lines [57].

### 5.2. AI in HCC Early Diagnosis

#### 5.2.1. AI in Imaging Diagnosis

The application of AI in medical imaging has brought increased convenience to the diagnosis of diseases and can assist in the analysis of imaging features.

Ultrasonography (US) is a commonly used method for screening for HCC; however, it is relatively less sensitive, with a sensitivity of only 46–63%, and is dependent on operator experience, equipment quality, and patient body habitus. Yang et al. (2020) developed a deep convolutional neural network (DCNN) using a large, multicenter ultrasound imaging database from 13 hospitals. The accuracy of this model for lesions detected by US was 86.5%, which was superior to that of contrast-enhanced CT (84.7%) and only slightly inferior to that of MRI (87.9%) [58]. Guo et al. demonstrated that a deep-learning (DL) algorithm applied to liver lesions seen by contrast-enhanced ultrasound (CEUS) could increase the sensitivity, specificity, and overall accuracy of CEUS for detecting HCC. It proved that the three-phase CEUS image-based, computer-aided diagnosis (CAD) is feasible for use with liver tumors, with the proposed deep canonical correlation analysis with multiple kernel learning (DCCA-MKL) framework [59].

Shukla et al. (2022) demonstrated an AI-driven, cascaded, fully convolutional neural network for HCC screening and prediction, based on various tested volumes of 3D-image reconstruction for comparison of algorithm database (3DIRCAD) datasets. This model showed that the total accuracy rate of the training and testing procedure was 93.85% [60]. AI and deep learning have revolutionized the traditional imaging diagnosis mode and have demonstrated high performance in the diagnosis and treatment of liver malignancies. Clinicians have used advanced deep-network algorithms to detect HCC lesions, diagnose diseases, and predict prognosis. It significantly enhances the efficiency of HCC diagnosis and treatment.

Mokrane and his colleagues conducted a small retrospective study on patients with liver cirrhosis and those with uncertain liver lesions, and they underwent diagnostic liver biopsy. Their radiomics signature was based on 13,920 CT imaging classifiers, the area under the receiver operating characteristic curve (AUC) of 0.70 (95%CI 0.61–0.80), and 0.66 (95%CI 0.64–0.84) in discovery and validation cohorts. The signature was influenced neither by segmentation nor by contrast enhancement [61].

To date, AI has been applied less frequently to MRI imaging of HCC than to CT. However, MRIs provide more data regarding liver imaging, and relying on the powerful computing power of AI, may have more promising prospects for assisting in the diagnosis of HCC and pose greater challenges for the design of AI models, with a sensitivity of 92%. Hamm et al. developed an NN algorithm with a convolutional neural network (CNN)-based deep-learning system (DLS), which classifies common hepatic lesions on multi-phasic MRI, successfully classifying MRI liver lesion specificity at 98% with an overall accuracy of 92% [62]. The above studies indicate that AI can enhance the decision-making ability of clinicians by identifying patients at high risk of HCC.

#### 5.2.2. AI in Histopathology

Pathological diagnosis is the gold standard for HCC diagnosis. Compared with imaging and other examination methods, it can more clearly describe the characteristics of the tumor and is of great significance for the formulation of treatment strategies [63]. Imaging is the primary method for screening typical HCC lesions based on imaging features. Although non-invasive criteria allow the diagnosis of HCC in most cases, histological examination of tumor samples is often required for masses with atypical features on imaging or for different diagnoses of benign liver tumors, cholangiocarcinoma (CCA; HCC in most cases), or metastasis. (Liao et al., 2020) used a CNN to distinguish HCC from adjacent normal tissues, using two large datasets of hematoxylin and eosin-stained digital slides, with an area under the curve (AUC) above 0.90 [64]. Kiana et al. (2020) developed a tool able to classify image patches as HCC or CCA, achieving an accuracy of 0.885 for the validation of the assisted state, and the model accuracy significantly affected the diagnostic decisions of all 11 pathologists [65]. AI can identify features that are beyond human eye recognition and relatively concealed, and quantitatively describe pathological details. It does not merely perform qualitative grading but provides a quantitative description of the pathological features. Moreover, the evaluation results are objective and consistent, avoiding diagnostic outcome variations caused by regional or subjective factors [66,67].

Many studies have confirmed that detection and diagnosis based on deep-learning algorithms have improved efficiency and achieved high diagnostic efficacy. This is an intelligent method for the detection and diagnosis of HCC.

### 5.3. AI in Drug Research and Development

Traditional drug development is time-consuming and expensive, typically commencing with the identification of a target and potentially spanning over a decade to unearth and refine a drug’s target of action. It also needs an additional period dedicated to development. As the pioneering and most advanced field within the -omics disciplines, multiomic data offers researchers a multifaceted lens into interconnected molecular insights, spanning static genomic details as well as dynamic, time- and space-sensitive expression and metabolic patterns. Unlike single-omic methods, integrated multiomic analysis provides a more holistic understanding of disease pathways, serving as a powerful tool for uncovering biomarkers and identifying potential therapeutic targets [68,69,70].

On 8 May 2024, AlphaFold 3 was published by Google DeepMind Technologies Limited, London UK. This model can be used in the accurate structure prediction of biomolecular interactions. Unlike earlier versions that required special treatment for different molecular types, AlphaFold 3’s framework aligns more closely with the fundamental physical principles governing molecular interactions. This makes the system significantly more efficient and reliable for studying novel molecular interactions [51,71,72]. AlphaFold 3’s sophisticated grasp of protein–ligand dynamics offers groundbreaking potential in the realm of drug discovery. Precisely pinpointing binding sites and determining the ideal configurations for drug candidates dramatically accelerates the drug design pipeline [73]. This capability is particularly transformative in the field of HCC research, where the rapid development of targeted therapies could yield treatments that are both more potent and less prone to adverse effects. Such advancements are reshaping therapeutic strategies and setting new benchmarks in precision medicine [74,75]. While challenges remain, such as the limited accuracy in predicting protein–RNA interactions, these hurdles are being addressed, paving the way for innovations in personalized medicine. AlphaFold still has many limitations, such as the restriction of static structure, the influence of chemical modification, and the impact of micro-environmental factors, including intracellular pH and ion concentration. The accuracy of functional prediction needs to be further improved [73]. Meanwhile, AI has cemented its role as a game-changer in drug development, revolutionizing the identification of new drug targets and repurposing of existing compounds, thereby opening new frontiers in medical research.

AI is also powerful in individualized medical support, such as HCC risk grade, prognosis, recurrence risk, toxicity, intervention path planning, and the optimization of surgery and treatment [5,76].

## 6. Conclusions

The HCC landscape is evolving rapidly with advances in biomarkers and multimodal therapies. Although challenges such as drug resistance and accessibility persist, the integration of immunotherapy and precision oncology heralds a new era in HCC management.

The increasing number of deaths caused by HCC is a growing concern. It is hoped that universal HBV vaccination, rising cure rates of HCV infection, and improved monitoring will alleviate this burden. HCC is a complex disease often associated with liver cirrhosis, and a multidisciplinary approach in specialized clinics is needed to maximize impact on the disease course. In the past decade, clinical management of HCC has improved, especially in patients with advanced HCC. Other areas of management still lack effective interventions, such as chemoprevention in patients with liver cirrhosis and adjuvant therapy after surgical resection or ablation. As the number of effective systemic drugs discovered in phase 3 trials continues to grow, the challenge lies in determining the sequence of systemic treatments to maximize clinical benefits while achieving the least toxic effects and costs. In the early stages, the prospects of combination therapy and the use of systemic drugs will shape future research plans for HCC.

## Figures and Tables

**Figure 1 biomedicines-13-01244-f001:**
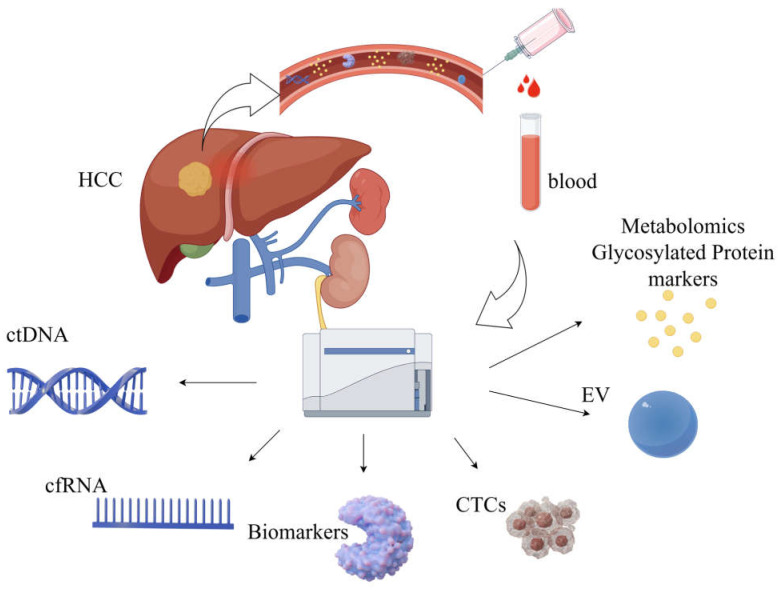
HCC biomarkers include circulating tumor DNA (ctDNA), cfRNA (cell-free RNA), seromarkers, circulating tumor cells (CTCs), extracellular vesicles (EV), and metabolomic markers.

**Figure 2 biomedicines-13-01244-f002:**
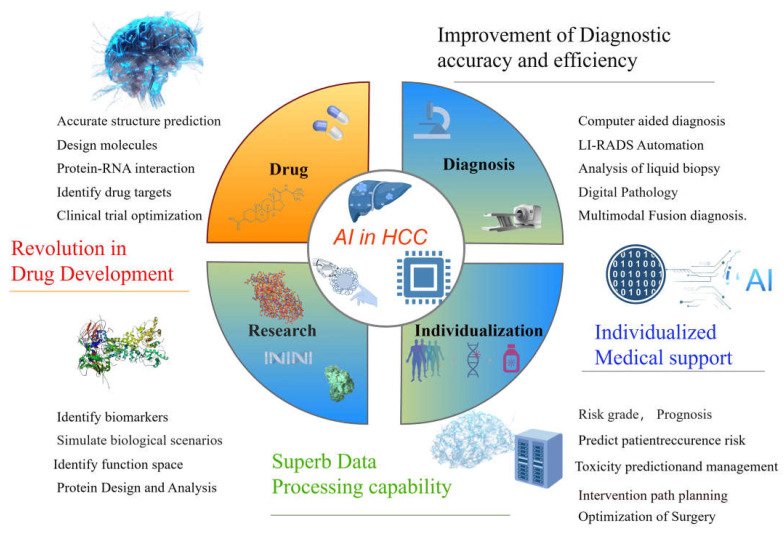
The application of AI to HCC, including drug development, early diagnosis, pathophysiological research, and individualized medical support.

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
