# Peer review of "New Insights into the Diagnosis and Treatment of Hepatocellular Carcinoma"

_biomedicines, 2025, doi:10.3390/biomedicines13051244_

Round 1

Reviewer 1 Report

Comments and Suggestions for Authors

In this narrative review the authors present current literature
on New Insights into the Diagnosis and Treatment of Hepatocellular Carcinoma. The review covers a broad overview of HCC pathophysiology, current therapies and the potential use of AI in the future treatment of the disease. 

The review is generally well structured and includes relevant references. Some sections do not flow well and could benefit from rewriting.  There are sufficient recent citations included in the text and no obvious omissions of relevant citations.

Figure 1 concept is good but make labels more clear and add more detailed description to legend. Currently the fonts look different. 

Figure 2:  more detailed description to legend. Currently the fonts look different. 

Line 36 : change boosting> improving

Line 40-41: think genome should be genomic 

46: change to "a glimmer"

47: witnessed is not the correct word, maybe undergone

59: change to "is the dynamic "

66: include additional references, eg for oncofetal genes. dont think cancer antigen genes is the correct wording, 

68: change to neoantigens 

82-83: explain why IFNg levels are relavant, eg immune activation 

92: change to "can be used"

94-98: reword for clarity 

109-110: words missing, reword for clarity 

114: have other groups also seen this? add additional references. change wording 

128 and 130: abbreviations are used that have previously been defined. Can just use the abreviations 

145: typo

152: reword for clarity 

162: what is meant by "this scientific group" name the author 

173: change to "can also"

172-175: add more detail, very brief doesnt add anything 

195: change certain > some 

208: change to cancer stem cells 

134: enabled>enable. use present tense 

260: contain>containing 

279-280: reword for clarity 

311: undergo> underwent 

312: typo

316-319: too long, break into shorter sentences. 

359: reword for clarity 

395-396: reword for clarity 

Please correct minor typos, as I have mentioned above. Check for spacing and use of capital letters.

Author Response

Thank you very much for taking the time to review this manuscript. Thank you very much for your valuable suggestions.

I have provide provide a point-by-point response to your suggestions.

Response to Reviewer 1 Comments

1. Summary

Thank you very much for taking the time to review this manuscript. Thank you very much for your valuable suggestions.

2. Point-by-point response to Comments and Suggestions for Authors

Comments 1: Figure 1 concept is good but make labels more clear and add more detailed description to legend. Currently the fonts look different.

Response : Figure 1 concept is good but make labels more clear and add more detailed description to legend. Currently the fonts look different. Thank you for pointing this out. I/We agree with this comment. Therefore, Ihave changed it

Comments 2: Figure 2:  more detailed description to legend. Currently the fonts look different. 

Response : Agree. I have modified figure2

Comments 3Line 40-41: think genome should be genomic 

Response :    Agree. I have modified

Comments Line 36 : change boosting> improving

Response Agree. I have modified

Comments 46 change to "a glimmer"

Response Agree. I have modified

Comments 47: witnessed is not the correct word, maybe undergone

Response Agree. I have modified

Comments 59: change to "is the dynamic "

Response Agree. I have modified

Comments 66: include additional references, eg for oncofetal genes. dont think cancer antigen genes is the correct wording, 

Response Agree. I have amodified

Comments 68: change to neoantigens 

Response I have modified

Comments 82-83: explain why IFNg levels are relavant, eg immune activation 

Response I insert the effect of interferonγand explain why HCC inhibit interferonγ

Comments 92: change to "can be used"

Response I have modified can be used"

Comments 94-98: reword for clarity 

Response I have  reword  it

Comments 109-110: words missing, reword for clarity 

Response I have modified

Comments114: have other groups also seen this? add additional references. change wording 

Response  The meaning of this passage is that the limitation of AFP is that in both early-stage and advanced-stage HCC, there are certain proportions of patients whose AFP levels are normal.

Comments128 and 130: abbreviations are used that have previously been defined. Can just use the abreviations 

Response It has been changed to an abbreviation.

Comments145 typo

Response I have modified

Comments152: reword for clarity 

Response I have reword it

Comments162: what is meant by "this scientific group" name the author 

Response mean Roskams-Hieter and his colleague

Comments173: change to "can also"

Response I have modified

Comments172-175: add more detail, very brief doesnt add anything 

Response I have insert detail

Comments195: change certain > some 

Response  I have modified certain in 200

Comments208: change to cancer stem cells 

Response I have modified

Comments134: enabled>enable. use present tense 

Response  234? enabled>enable

Comments260: contain>containing 

Response  I have modified

Comments279-280: reword for clarity 

Response I have Change it to a more understandable expression.

Comments311: undergo> underwent 

Response I have modified

Comments312: typo

Response I have modified

Comments316-319: too long, break into shorter sentences. 

Response  I have modified

Comments359: reword for clarity 

Response  I have modified

Comments395-396: reword for clarity 

Response  I have modified

CommentsPlease correct minor typos, as I have mentioned above. Check for spacing and use of capital letters.

Response  I have modified

Comments

Response

Reviewer 2 Report

Comments and Suggestions for Authors

The authors attempt to cover a large topic comprehensively. Although the effort is admirable, I feel the organization of the paper could be restructured to improve readability. In addition there are some content areas that should be filled out more. As the authors provide a detailed focus on AI, i would suggest the the last paragraph in section 3.1 could be eliminated as AI and imaging is discussed elsewhere. In this same section, the authors mention MRI imaging but do not discuss CT or CEUS although they discuss these modalities in the AI sections. The imaging section should have discussion of CT and US as screening modalities as MRI is not universally available. The section should also reference guidelines for screening (who, how often etc.). I

In section 4.1, references to studies using neoadjuvant and adjuvant immunotherapy with surgical intervention are not mentioned. Its is important to mention these trials as they represent a new fronteir in multimodality therapy for treatment. Further in the interventional section, histotripsy is also not mentioned and should be referenced.

Author Response

Thank you very much for taking the time to review this manuscript. Thank you very much for your valuable suggestions.

 The point-by-point response to you is in the word .

Response to Reviewer 1 Comments

1. Summary

Thank you very much for taking the time to review this manuscript. Thank you very much for your valuable suggestions.

2. Point-by-point response to Comments and Suggestions for Authors

Comments 1: The authors attempt to cover a large topic comprehensively. Although the effort is admirable, I feel the organization of the paper could be restructured to improve readability. In addition there are some content areas that should be filled out more. As the authors provide a detailed focus on AI, i would suggest the the last paragraph in section 3.1 could be eliminated as AI and imaging is discussed elsewhere.

Response :I will delete he last paragraph in section 3.1

Comments

 In this same section, the authors mention MRI imaging but do not discuss CT or CEUS although they discuss these modalities in the AI sections. The imaging section should have discussion of CT and US as screening modalities as MRI is not universally available. The section should also reference guidelines for screening (who, how often etc.).

Response :I will make the necessary revisions according to your suggestions.

Comments

In section 4.1, references to studies using neoadjuvant and adjuvant immunotherapy with surgical intervention are not mentioned. Its is important to mention these trials as they represent a new fronteir in multimodality therapy for treatment. Further in the interventional section, histotripsy is also not mentioned and should be referenced.

Response :   I will make the necessary revisions according to your suggestions.Add the following content to the translation:neoadjuvant and adjuvant immunotherapy with surgical interventionï¼›histotripsy。

Reviewer 3 Report

Comments and Suggestions for Authors

This manuscript goes into the complexities of diagnosing and treating hepatocellular carcinoma, which is a major cause of cancer death around the globe, often linked to chronic liver ailments like hepatitis and fatty liver disease. While antiviral treatments are improving the approach to infections, the increase in cases because of non-infectious reasons highlights an urgent need for better diagnostic tools and treatment options. It covers advancements in imaging, liquid biopsies, diverse therapeutic strategies, and the role of artificial intelligence in diagnosis and treatment.

The manuscript provides a review of hepatocellular carcinoma (HCC), including its epidemiological context, pathogenesis, diagnostic methods, and treatment strategies. It explores the role of artificial intelligence (AI) in diagnosis and treatment development, highlighting its potential to improve accuracy and efficiency. The manuscript advocates a collaborative approach involving various specialties, which aligns with current trends in personalized medicine and can improve clinical outcomes. It highlights the importance of early diagnosis and intervention, which are essential to improve survival rates among HCC patients, and discusses novel biomarkers and liquid biopsy technologies, an important aspect, suggesting advances that could lead to more effective diagnostic tools.

Timely and accurate diagnosis of HCC remains a challenge, particularly in resource-limited settings. The high cost and length of the current HCC testing process frequently cause patients to delay treatment, resulting in late-stage cancer diagnoses that prevent surgery. Therefore, the authors should have primarily answered this question: how does artificial intelligence affect the treatment of hepatocellular carcinoma and what advances do its findings help in early diagnosis?

Although the potential of AI is substantial, the authors show a potential overemphasis on AI. This emphasis could lead to neglecting other important factors in the management of HCC, such as traditional clinical methods and patient disparities. The manuscript needs a more thorough discussion of limitations. There is a lack of in-depth discussion of the limitations of current diagnostic and treatment modalities, such as challenges in access to care or socioeconomic factors that influence patient outcomes. AI doesn't seem to do much about prevention.

In fact, what leaves me perplexed is that HCC is a sneaky disease that does not show symptoms until it is very advanced. Doctors usually discover HCC in most patients only during emergency room visits. The manuscript emphasizes new technologies and even AI, but these are analyses to be used on people who already have symptoms, not for prevention.

The point of greatest interest concerns the stealthy nature of hepatocellular carcinoma (HCC) and the challenges associated with its early diagnosis. HCC often develops silently, leading to a diagnosis at advanced stages. This highlights a significant gap in the current strategies focused on detection and prevention, which causes patients to die because the cancer is already at an advanced stage, making surgery unuseful.

We should consider the need to improve screening practices for at-risk populations before symptoms appear. Even the regular screening based on risk factors (e.g., chronic hepatitis B or C infection, cirrhosis) is crucial for early detection but is often underutilized in practice. While the manuscript discusses advancements in imaging and AI, it is crucial for these technologies to be integrated into preventive strategies that go beyond symptomatic patients. Ideally, they should be part of routine screening programs aimed at identifying pre-symptomatic or asymptomatic cases, which can significantly improve outcomes through earlier intervention. Therefore, the manuscript could benefit from a stronger emphasis on primary prevention strategies. There should be a rational discussion on establishing guidelines for screening high-risk populations using advanced technologies and biomarkers. This would enhance the manuscript's message by connecting new technological capabilities to practical applications in early detection and prevention efforts.

Raising the awareness among patients and healthcare providers about the risks associated with liver disease and the potential for HCC formation can drive proactive health behaviors and encourage earlier consultations, thus facilitating earlier detection. While advancements in technology and AI are promising for improving HCC diagnosis and treatment, a more focused approach to prevention and routine screening practices for at-risk individuals is essential. Addressing these gaps helps shift the narrative from reactive to proactive management of HCC, ultimately improving survival rates and patient outcomes.

 The inadequacy of new technologies in detecting and managing hepatocellular carcinoma (HCC) without accompanying prevention programs can have several implications. Technologies like advanced imaging (CT, MRI) and AI-driven diagnostic tools enhance detection in patients who already exhibit symptoms or signs of disease. If we use these technologies primarily reactively, they will probably cannot diagnose HCC at an earlier, more treatable stage, thus preventing interventions that could significantly improve survival rates.

Doctors diagnose many patients only after they present with severe symptoms or complications. This late-stage diagnosis limits treatment options and reduces the likelihood of successful outcomes. Using advanced technologies in emergency settings can lead to higher healthcare costs because of hospitalizations, emergency care, and treatment for advanced disease. Without prevention programs to reduce the incidence of HCC, healthcare resources can become strained, leading to inefficiencies in the system.

Access to advanced diagnostic tools can vary significantly across different regions and populations, particularly in lower-income areas. Without a robust prevention strategy that focuses on education and screening of high-risk populations, disparities in HCC outcomes will probably persist or widen.

Relying on diagnostic technologies without addressing the underlying causes of HCC, such as viral hepatitis, alcohol abuse, and the deep molecular mechanisms that manifest as metabolic alterations, means that new technologies alone cannot effectively reduce the incidence of fresh cases. Unfortunately, without an understanding of the deep biological processes of HCC, we will never have a cure. This is clear to everyone.

Prevention programs that focus on vaccination, risk education, and lifestyle changes are essential. The perception of new technologies as panaceas for HCC detection might give healthcare providers and patients a false sense of security. This might mean a less focus on preventing illness, which could lead to more cases of disease.

Finally, advanced imaging and AI tools come with challenges, including false positives or negatives. It has already happened in the USA for breast cancer in black women. Without robust screening programs and clinical guidelines, there might be inconsistent interpretations of diagnostic results leading to either unnecessary interventions or missed opportunities for treatment. Without an integrated approach that emphasizes both prevention and early detection, the public health burden of HCC will probably continue to grow. Comprehensive strategies encompassing vaccination campaigns, screening at-risk individuals, and community health initiatives are critical.

The crucial need for new technologies and artificial intelligence is not only to improve diagnostic details in symptomatic patients but to discover fresh cases in time with the lowest cost.

Author Response

Thank you very much for taking the time to review this manuscript. Thank you very much for your valuable suggestions.

The point-by-point response is in the word.

Response to Reviewer 3 Comments

1. Summary

Thank you very much for taking the time to review this manuscript. Thank you very much for your valuable suggestions.

2. Point-by-point response to Comments and Suggestions for Authors

Comments 1: This manuscript goes into the complexities of diagnosing and treating hepatocellular carcinoma, which is a major cause of cancer death around the globe, often linked to chronic liver ailments like hepatitis and fatty liver disease. While antiviral treatments are improving the approach to infections, the increase in cases because of non-infectious reasons highlights an urgent need for better diagnostic tools and treatment options. It covers advancements in imaging, liquid biopsies, diverse therapeutic strategies, and the role of artificial intelligence in diagnosis and treatment.

The manuscript provides a review of hepatocellular carcinoma (HCC), including its epidemiological context, pathogenesis, diagnostic methods, and treatment strategies. It explores the role of artificial intelligence (AI) in diagnosis and treatment development, highlighting its potential to improve accuracy and efficiency. The manuscript advocates a collaborative approach involving various specialties, which aligns with current trends in personalized medicine and can improve clinical outcomes. It highlights the importance of early diagnosis and intervention, which are essential to improve survival rates among HCC patients, and discusses novel biomarkers and liquid biopsy technologies, an important aspect, suggesting advances that could lead to more effective diagnostic tools.

Timely and accurate diagnosis of HCC remains a challenge, particularly in resource-limited settings. The high cost and length of the current HCC testing process frequently cause patients to delay treatment, resulting in late-stage cancer diagnoses that prevent surgery. Therefore, the authors should have primarily answered this question: how does artificial intelligence affect the treatment of hepatocellular carcinoma and what advances do its findings help in early diagnosis?

Response :

Baocheng Deng  have already written a review.

Hepatocellular carcinoma: molecular mechanism, targeted therapy, and biomarkers Cancer Metastasis Rev. 2023 Sep;42(3):629-652. doi:10.1007/s10555-023-10084-4.

This review system descript Biomarker;Mechanism; Molecular target; Signaling pathway of HCC.Over the past two years, the revolutionary progress has been rather limited. Therefore, this article focuses more on the assistance provided by AI in HCC's pathological physiology research, drug development, and diagnostic techniques.

Based on your suggestion, I will insert some research related to the early diagnosis assistance provided by AI.

Comments :Although the potential of AI is substantial, the authors show a potential overemphasis on AI. This emphasis could lead to neglecting other important factors in the management of HCC, such as traditional clinical methods and patient disparities. The manuscript needs a more thorough discussion of limitations. There is a lack of in-depth discussion of the limitations of current diagnostic and treatment modalities, such as challenges in access to care or socioeconomic factors that influence patient outcomes. AI doesn't seem to do much about prevention.

In fact, what leaves me perplexed is that HCC is a sneaky disease that does not show symptoms until it is very advanced. Doctors usually discover HCC in most patients only during emergency room visits. The manuscript emphasizes new technologies and even AI, but these are analyses to be used on people who already have symptoms, not for prevention.

Response : HCC is a sneaky disease that does not show symptoms until it is very advanced.However, for high-risk groups, when regular screening is conducted, new technologies can indeed increase the probability of early detection. Based on this understanding, any means that can help improve the rate of early detection should be given attention.Due to the limitations of the article's length, there are indeed certain restrictions. It merely reflects a difference in the focus of the writing.

Comments :The point of greatest interest concerns the stealthy nature of hepatocellular carcinoma (HCC) and the challenges associated with its early diagnosis. HCC often develops silently, leading to a diagnosis at advanced stages. This highlights a significant gap in the current strategies focused on detection and prevention, which causes patients to die because the cancer is already at an advanced stage, making surgery unuseful.

Response : Yes, this challenge of HCC was already raised in the opening part of the article. This article merely summarizes some technological advancements that might enhance the survival rate of patients, with a different focus.

Comments :We should consider the need to improve screening practices for at-risk populations before symptoms appear. Even the regular screening based on risk factors (e.g., chronic hepatitis B or C infection, cirrhosis) is crucial for early detection but is often underutilized in practice. While the manuscript discusses advancements in imaging and AI, it is crucial for these technologies to be integrated into preventive strategies that go beyond symptomatic patients. Ideally, they should be part of routine screening programs aimed at identifying pre-symptomatic or asymptomatic cases, which can significantly improve outcomes through earlier intervention. Therefore, the manuscript could benefit from a stronger emphasis on primary prevention strategies. There should be a rational discussion on establishing guidelines for screening high-risk populations using advanced technologies and biomarkers. This would enhance the manuscript's message by connecting new technological capabilities to practical applications in early detection and prevention efforts.

Response :Yes, research on AI-assisted examination to enhance the probability of early diagnosis is still at a very rudimentary stage. However, it is also true that the advantages brought by powerful computing power should be recognized. Even though this computing power is still some distance away from clinical application, it should not be ignored.

Comments Raising the awareness among patients and healthcare providers about the risks associated with liver disease and the potential for HCC formation can drive proactive health behaviors and encourage earlier consultations, thus facilitating earlier detection. While advancements in technology and AI are promising for improving HCC diagnosis and treatment, a more focused approach to prevention and routine screening practices for at-risk individuals is essential. Addressing these gaps helps shift the narrative from reactive to proactive management of HCC, ultimately improving survival rates and patient outcomes.

Response :I fully agree with your point of view. Your view is completely not contradictory to the possible technological progress that this article intends to summarize.

Comments  The inadequacy of new technologies in detecting and managing hepatocellular carcinoma (HCC) without accompanying prevention programs can have several implications. Technologies like advanced imaging (CT, MRI) and AI-driven diagnostic tools enhance detection in patients who already exhibit symptoms or signs of disease. If we use these technologies primarily reactively, they will probably cannot diagnose HCC at an earlier, more treatable stage, thus preventing interventions that could significantly improve survival rates.

Response :At present, the related technologies are only being attempted for application in clinical practice. The previously mature technologies have not been eliminated. Any technology that is beneficial to patients can and should be applied, as long as it does not cause harm to patients.

Comments Doctors diagnose many patients only after they present with severe symptoms or complications. This late-stage diagnosis limits treatment options and reduces the likelihood of successful outcomes. Using advanced technologies in emergency settings can lead to higher healthcare costs because of hospitalizations, emergency care, and treatment for advanced disease. Without prevention programs to reduce the incidence of HCC, healthcare resources can become strained, leading to inefficiencies in the system.

Response :Yes, Prevention measures are of great significance for HCC. However, the focus of this article lies in envisioning some potential technological advancements that may benefit patients, without denying the role of prevention in HCC.

Comments Access to advanced diagnostic tools can vary significantly across different regions and populations, particularly in lower-income areas. Without a robust prevention strategy that focuses on education and screening of high-risk populations, disparities in HCC outcomes will probably persist or widen.

Response :During the initial stage of the market launch of advanced diagnosis and treatment technologies, the investment was indeed huge. However, as the application expands, in terms of market competition, the cost will significantly decrease. Eventually, even low-income groups will benefit. These technologies do initially benefit high-income groups, but in the end, low-income groups will also benefit. Of course, it is not denied that education and preventive measures for high-risk groups in low-income areas are of great importance.

Comments Relying on diagnostic technologies without addressing the underlying causes of HCC, such as viral hepatitis, alcohol abuse, and the deep molecular mechanisms that manifest as metabolic alterations, means that new technologies alone cannot effectively reduce the incidence of fresh cases. Unfortunately, without an understanding of the deep biological processes of HCC, we will never have a cure. This is clear to everyone.

Response :Yes, the advancement of diagnostic techniques cannot completely replace the role in understanding the pathogenesis and prevention of diseases. However, at the present stage, diagnostic techniques are an important aspect in improving the survival time of patients.

Comments :Prevention programs that focus on vaccination, risk education, and lifestyle changes are essential. The perception of new technologies as panaceas for HCC detection might give healthcare providers and patients a false sense of security. This might mean a less focus on preventing illness, which could lead to more cases of disease.

Response :Yes, prevention is the cornerstone of disease control. However, it is undeniable that liver cancer will still exist for a long time in human society. This is an undeniable fact in a specific historical period. Based on this fact, it is also very important to conduct secondary prevention for people with existing risk factors and to provide treatment for those who have already developed liver cancer.Primary prevention aims to reduce the occurrence of diseases; secondary prevention is to lower the incidence of advanced cases; tertiary prevention is to improve the quality of life of patients.both of them are very important.

Comments Finally, advanced imaging and AI tools come with challenges, including false positives or negatives. It has already happened in the USA for breast cancer in black women. Without robust screening programs and clinical guidelines, there might be inconsistent interpretations of diagnostic results leading to either unnecessary interventions or missed opportunities for treatment. Without an integrated approach that emphasizes both prevention and early detection, the public health burden of HCC will probably continue to grow. Comprehensive strategies encompassing vaccination campaigns, screening at-risk individuals, and community health initiatives are critical.

Response :Yes, every technology has its specific limitations. Even for doctors, it's no different. The accuracy cannot reach 100%. If there are data available to prove that these technological advancements can enhance the diagnostic accuracy rate, they should be attempted to be applied. Although there might be some legal and other related issues to deal with.

Comments The crucial need for new technologies and artificial intelligence is not only to improve diagnostic details in symptomatic patients but to discover fresh cases in time with the lowest cost.

Response :Tertiary prevention is a stratified prevention strategy adopted in public health and clinical medicine for different stages of diseases, aiming to reduce the occurrence, development and disability risks of diseases.It is never denied that low-cost measures such as primary prevention are effective. However, the reality is that for a certain period of time, the significant role of advancements in treatment and diagnostic technologies has been important.

Round 2

Reviewer 3 Report

Comments and Suggestions for Authors

see attached pdf

Author Response

I express my sincere gratitude to you for your valuable suggestions and the precious time you have devoted. I also hope you can understand that the focus of this article is on the progress of computer technology and algorithms, which has brought significant help to the diagnosis and treatment of diseases. Computers and algorithms have indeed enhanced our work efficiency, allowing us to have more energy to do things beneficial to patients.

Please refer to the attachment for details.

Round 3

Reviewer 3 Report

Comments and Suggestions for Authors

The authors show an emphasis on AI and computational methods within the manuscript that reflects the broader trend in the scientific community where advanced computational techniques are increasingly viewed as essential tools for addressing complex biomedical problems. AI can process and analyze vast datasets generated from high-throughput experiments enabling researchers to identify patterns and make predictions.

But there is a fundamental condition that underlies any use of AI.

Artificial intelligence and its algorithms work on data produced by machines that make measurements transformed into data with the help of mathematical criticism. If the data are a direct expression of the measurements, AI is very useful, if the data are an indirect expression (there are no machine measurements), AI is useless because it would produce incorrect and untrue evaluations.

The basic crucial aspect is the reliance on accurate, machine-measured data.

AI indeed shines when working with datasets that are the direct output of reliable and precise measurements. In such cases, it can extract patterns, predict outcomes, and reveal connections that may be impossible for humans to discern manually. However, when data lacks a robust foundation—such as being indirectly derived or based on assumptions—AI is at risk of amplifying inaccuracies. This can lead to flawed predictions and decisions that may not hold up in practical applications. Moreover, this principle highlights the importance of data provenance and validation. Before feeding data into AI systems, researchers must critically assess its source, method of collection, and relevance to the question at hand. The value of artificial intelligence lies not in its analytical capacity, but in the quality of the data it processes.

The use of AI in medicine is both transformative and deeply thought-provoking. On one hand, AI's ability to analyze vast datasets with precision can lead to incredible advancements—like identifying patterns in genetic data, predicting disease outcomes, or optimizing treatment plans tailored to individual patients. This can improve diagnostic accuracy, enable earlier interventions, and enhance overall patient care.

However, the reliability of AI's assessments hinges on the quality and nature of the data it's trained on. In medicine, this becomes critically important because decisions based on flawed or incomplete data can have real consequences for patients' health. Moreover, indirect data or assumptions could lead AI systems to provide misleading predictions, potentially putting patients at risk. AI has immense potential to revolutionize medicine, but it must be integrated thoughtfully, with human oversight and ethical frameworks guiding its use.

Hepatocellular carcinoma (HCC) and the challenges of applying AI to such a complex and poorly understood disease are of great relevance. The limited dataset of experimentally validated protein interactions (less than 5%) represents a challenge, but it represents a formidable limitation, today insurmountable.

While many of these methodologies hold promise, it's crucial to address challenges such as data standardization, model explainability, and the integration of AI insights into clinical practice. Collaboration between AI researchers, clinicians, and biologists will be key to overcoming these hurdles.

Enriching sparse interaction data with data from predictive models is not a good system. We must not forget that algorithms that predict protein interactions today are forced to use a database where less than 5% of the data that make sense for AI. Furthermore, we forget that AlphaFold static models are proteins, which are very mobile biological objects, and that mobility controls both interaction and function. Therefore, the static models generated are useless for designing new drugs that must interact with a mobile part of the protein object. The results are a waste of time, because they will be the same as those obtained without AI with old docking methodologies. Where is the improvement in AI? The same goes for biomarkers. If you don't have good interactomics, you can't know the causal molecular mechanisms. If you don't know the causal mechanisms, you can't know who can serve as a biomarker. If there are no specific and effective treatments, how can you design clinical approaches supported by little or nothing?

So we must conclude that these AI-supported methodologies are promising, they should be implemented but only if there are the conditions of knowledge based on certain, valid and reliable data sets for at least  70 – 75%. The remaining data can also be replenished with indirect data. The real problem is man and his ignorance, who, not knowing that he does not know, thinks he knows more than others.

AI can process and analyze vast datasets generated from high-throughput experiments enabling researchers to identify patterns and make predictions.

But it remains the fundamental condition underlying any use of AI.

Artificial intelligence and its algorithms work on data produced by machines that make measurements transformed into data with the help of mathematical criticism.

If the data are a direct expression of the measurements, AI is very useful; if the data are an indirect expression (there are no machine measurements), AI is useless because it would produce incorrect and untrue assessments.

AI indeed shines when working with datasets that are the direct output of reliable and precise measurements. In such cases, it can extract patterns, predict outcomes, and reveal connections that may be impossible for humans to discern manually. However, when data lacks a robust foundation—such as being indirectly derived or based on assumptions—AI is at risk of amplifying inaccuracies. This can lead to flawed predictions and decisions that may not hold up in practical applications.

Moreover, this principle highlights the importance of data provenance and validation. Before feeding data into AI systems, researchers must critically assess its source, method of collection, and relevance to the question at hand. AI's value lies not only in its analytical capability but also in the quality of the data it processes.

This my note also raises questions about the limitations of AI and the human oversight required in interpreting its results. Machines might process data, but humans bear the responsibility of ensuring its integrity. This critique raises fundamental issues about the limitations of current AI methodologies in the context of complex biological systems like hepatocellular carcinoma (HCC).

Regarding clinical approaches, the foundation of any AI-supported methodology in medicine must be the existence of reliable, comprehensive datasets. As noted, if we aim for datasets that are only partially validated (at least around 70-75%), AI may still serve to fill gaps intelligently. But anything less than that threshold risks leading to pseudoscientific conclusions rather than meaningful breakthroughs.

AI is an incredible tool, but it operates within the boundaries of human knowledge. It doesn't transcend ignorance; it merely reflects it. This underscores the importance of researchers embracing humility, investing in data validation, and understanding the limitations of the tools they use.

Generally, human knowledge today proceeds quickly on systems, such as biomedical ones, which are complex by definition. To better organize critical collective knowledge, AI is not enough, it is first necessary to establish a multidisciplinarity that requires knowledge of a common language. Today, bridge scientists are very few. Without glue, a multidisciplinary group cannot be created. Without a multidisciplinary clinical group, you cannot get anywhere, especially in the case of a disease whose basic knowledge is based on only 5% of data that can be effectively used by AI.

The authors cannot ignore these fundamental aspects at this moment in human knowledge. This is a review, and these aspects cannot be omitted. The reader must know to understand and cannot be kept in the dark with emphatic tones.

The authors, whether they are technicians or clinicians, should include a paragraph that clearly explains the general problem and the implications on HCC that are quite heavy and serious.